# Study on the Root Causes and Prevention of Coating Cracks in the Cargo Hold of a Product Carrier

**Myung-Su Yi** [1], **Kwang-Cheol Seo** [2] **and Joo-Shin Park** [3,*]

1 Department of Naval Architecture and Ocean Engineering, Chosun University, Gwangju 61452, Korea
2 Department of Ocean System Engineering, Graduate School, Mokpo National Maritime University, Mokpo 58628, Korea
3 Ship & Offshore Research Institute, Samsung Heavy Industries Co., Ltd., Geoje 53261, Korea
* Correspondence: scv7076@nate.com; Tel.: +82-55-630-9613

**Abstract:** Recently, shipyards have been booming in the product carrier (PC) market of the global shipbuilding industry. Due to the rising market conditions, orders for large container vessels and PCs are also steadily increasing. According to the industry, freight rates for LR tankers (70K) and MR tankers (50K) in the shipbuilding industry have rose to the highest level during the previous year. In order to secure the competitiveness of PCs, various core items are required, and Korean shipbuilders have been leading the market for a long time based on their knowledge of the design and production of low-fuel ships. In recent years, there have been frequent cases of coating cracks in the cargo hold after sea trial. All relevant rules presented by the classification so far are structural design, safety evaluation, and inspection standards, and coating cracks are considered a problem for coating makers. In other words, they must establish a standard coating measure agreed upon by all parties (owner, shipbuilder, and coating maker); therefore, solutions have been proposed by each shipping company. In this study, the causes of coating cracks occurring in the cargo hold of PCs during tank tests in sea trials were analyzed and measures to prevent them were studied. The main point of this study is assumed that coating crack are caused by mechanical load induced by structural behavior, and the numerical analysis methodology is newly introduced. Based on the results of the numerical analysis, it was confirmed that there is a high probability of coating cracks in the critical area where high stress occurs in the cargo hold. Therefore, the results obtained in this study will be useful to prevent coating cracks in future PC designs.

**Keywords:** PC (product carrier); coating crack; corrosion; tank test; hold; sea trial

## 1. Introduction

Vessel orders stagnated due to a decrease in global cargo volume influenced by COVID-19 pandemic. However, domestic exports of ships also reached USD 23 billion, representing an increase of 16% compared to previous year (USD 19.7 billion), the highest performance in four years since 2017. The vessels receiving the most orders were LNG carriers and LNG-fueled container ships for environmental reasons. Furthermore, the number of vessel orders is continuously increasing due to the effect of increasing freight rates for product carriers. Due to the characteristics of product carriers transporting various liquid cargoes, contamination by impurities in cargo holds is strictly limited and managed. This is also why special coating specifications are applied to cargo holds, unlike typical tankers. The competitiveness of Japan and China in the global product carrier market lies in the design of lightweight structures and their application in eco-friendly, fuel-efficient hulls.

Recently, there has been an increase in the number of cases of coating cracks in cargo holds during tank tests in sea trials. When a coating crack occurs in the cargo hold, rust flows down, eventually contaminating the cargo. Therefore, ship owner are motivated to determine the root cause of this problem, as such contamination is a major source of dispute between owners and shipyard.

In this study, with the aim of determining the root cause of coating cracks, numerical cargo hold analysis was performed on critical areas generating large strains in tank tests using detailed FE modeling [1–3]. We calculated structural stresses according to loading conditions at various locations, including the cargo holds of product carriers. In general, the strain capacity of the coating is very low compared to that of the steel structure, and the main cause of coating cracking is stress of the structural framework. Therefore, finding and understanding the location where high stress occurs in the hull structure is the most important issue.

Previous relevant studies can be summarized as follows. C.P. Gardiner and R.E. Melchers [4] identified main types of corrosive environments, namely immersion in seawater, exposure to an enclosed atmosphere, and exposure to porous media in the cargo holds of bulk carriers. Based on the results of pattern study, it is clear that the parameters relating to the operation and design of bulk carriers influence the corrosion rate. Therefore, it was proposed that such parameters represent a good starting point for the development of a corrosion rate database. Ho Il Lee et al. [5] studied critical issues associated with the heating coil units of stainless steel and aluminum brass tube materials. Through case studies, the authors addressed, in detail, the cause of corrosion damage in heating coil units. Tatsuro Nakai et al. [6] observed pitting corrosion on hold frames in the cargo holds of bulk carriers that exclusively carry coal and iron. They demonstrated that the shape of the observed corrosion pits was circular and conical and that the ratio of the diameter to the depth ranged between 8 to 1 and 10 to 1, in contrast to the trend observed for the bottom shell oil tankers, for which the ratio ranges between 4 to 1 and 6 to 1. A series of tensile tests was conducted to investigate the effect of pitting corrosion on tensile strength. The reduction in tensile strength of the structure resulting from pitting corrosion is larger than that resulting from uniform thickness loss. Through buckling tests, it was found that buckling strength under pitting corrosion was lower than under uniform thickness loss of the plate. DNV [7] developed guidelines focusing on survey, maintenance, and repair procedures for tanker coatings to determine the level of coating repair required based data collected through ship inspections. Therefore, these guidelines are minimally relevant to the tank coating corrosion test investigated in the present study, as the data are obtained at the time of redocking. However, the data on various types of coating corrosion in tankers are well-organized, enabling even non-experts to decide whether the coating needs to be repaired. Do Kyun Kim et al. [8] focused on the historical trend of corrosion addition rules for ship structural design and investigated their effects on the ultimate strength performance of components such as hull girders and the stiffened panel of double-hull oil tankers. Five types of corrosion addition models were applied to investigate the general trend of corrosion addition. Based on the survey results, empirical formulae were proposed to determine the ultimate hull girder strength of double-hull oil tankers according to different corrosion addition rules. ABS [9] published guidance notes on coating maintenance and restoration of ballast tanks and cargo oil tanks in order to satisfy IMO-PSPC standards, at a minimum. The guidelines suggest that in order to maximize service life with reduced lifecycle cost, it is critical to maintain and repair the coating at an early stage before accelerated corrosion occurs due to coating deterioration. This guidance also focuses on the maintenance and repair of the coating from the point of view of classification societies, with no mention of coating cracks that occur during tank tests.

In this study, the causes of cracks occurring in the cargo hold of PCs during tank tests sea trials were analyzed, and measures to prevent such cracking were studied. Therefore, the results obtained in this study will be useful to prevent coating cracks in future PC designs.

## 2. Tank Test and Coating Crack Survey

### 2.1. Tank Test

Vessel sea trials newly built ships are important to ensure the overall safety of the ship, the crew, and the cargo. Among the tests conducted during sea trials, tank tests are

performed to confirm the water tightness of tanks, as well as their watertight boundaries and structural adequacy to ensure of the watertightness of ship subdivisions. The tightness of all tanks and watertight boundaries of ships during new construction, as well as those relevant to major conversions or major repairs, must be confirmed by these test procedures prior to delivery of the ship. Among the tests of the cargo hold, a tank test is performed to verify the structural adequacy of the tank construction. This test is usually performed as a hydrostatic test using seawater.

A typical product carrier and zig-zag loading scheme (hatching area) in the cargo hold during a tank test are shown Figure 1.

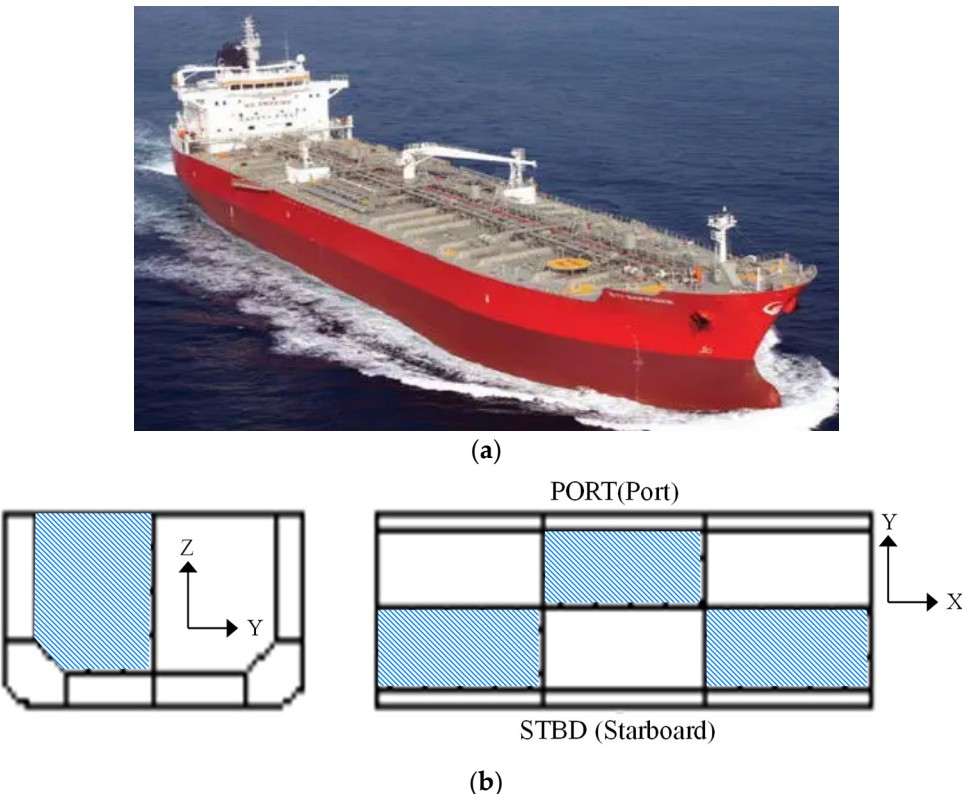

(a)

(b)

**Figure 1.** Typical product carrier and zig-zag loading pattern during tank test. (**a**) Typical product carrier; (**b**) zig-zag loading in the cargo hold.

The location of sea trials differs according to shipyard conditions, and the effect of hull girder bending induced by ship motion should be considered under real-world conditions, as shown Figure 2. The tank test is performed by additionally considering zig-zag cargo loading and the lateral pressure, as well as the draft.

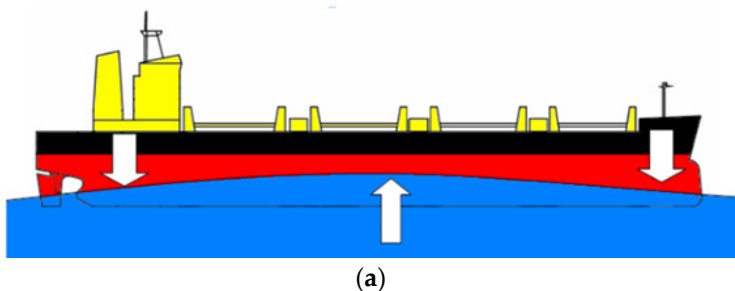

(a)

**Figure 2.** *Cont.*

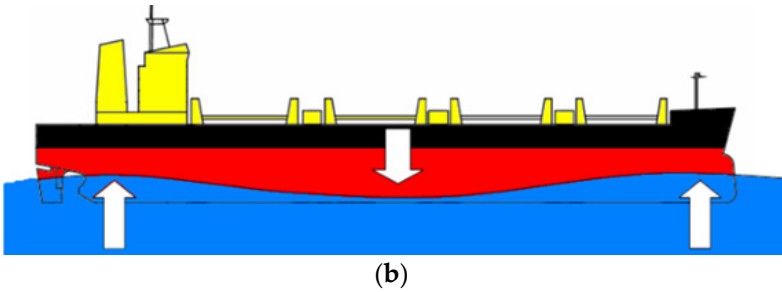

(**b**)

**Figure 2.** Typical hull girder loads: (**a**) hogging mode; (**b**) sagging mode.

*2.2. Coating Crack Survey*

According to the tank test, coating cracks occurred at various locations in the cargo hold. The most common locations are listed below, with examples shown in Figure 3.

- Ceiling at the connection between the center bulkhead and the main deck;
- Upper and lower stool and corner brackets; and
- Typical web welding toe.

These are the locations where structural stress is high during tank tests. Although the maximum stress is low, a coating crack may be generated. In most cases, impurities are added during coating, which may result in a thicker coating than that recommended by standard specifications. Basically, the tank test is conducted to evaluate the safety of structural members and is performed under more conservative conditions than the actual loading pattern. Therefore, coating cracking easily occurs under tank test conditions.

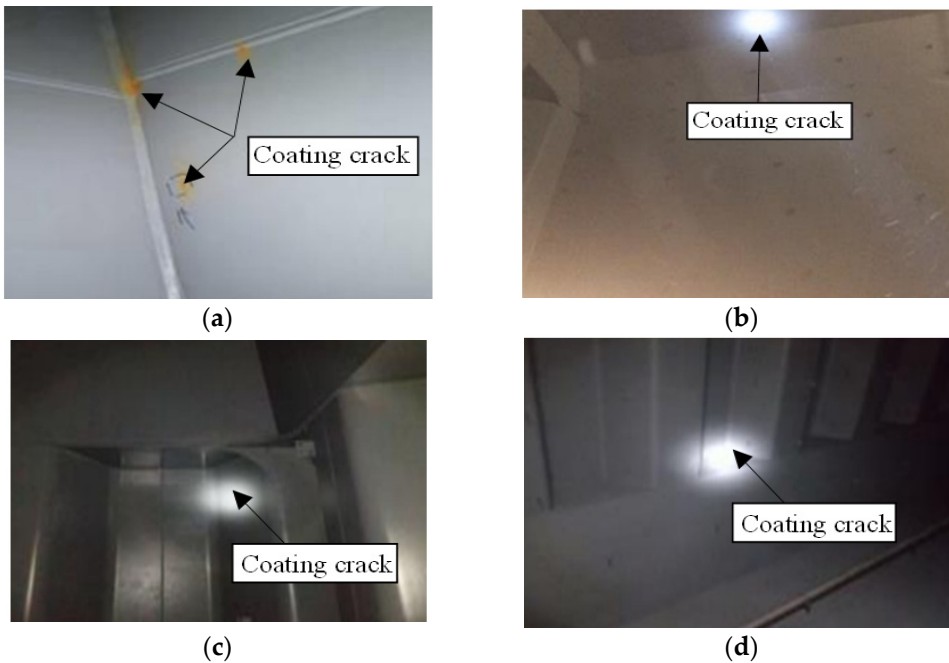

**Figure 3.** Coating crack survey results in the cargo hold: (**a**) low stool; (**b**) typical web section deck; (**c**) upper stool (bottom); (**d**) lower stool (bottom).

## 3. Analysis Method

*3.1. Analysis Procedure and Numerical Model*

Structural strength analysis was performed considering tank test conditions as recommended by HCSR (Harmonized Common Structural Rule) using a global FE model (so-called 1-longi. model), as shown Figure 4a. This analysis procedure was performed to confirm the location of maximum stress caused by the design loadings and allowable stress

considering the safety factor specified by classification societies using von Mises stress. The analysis procedure to check whether the coating is cracked is shown in Figure 4a; the coating is very thin, and the element is divided into 1.0 mm segments. The FE analysis involves calculation of nominal stress according to the tank test conditions and converted into plastic strain using Neuber's rule. If the plastic strain is larger than the rupture strain of the coating, then coating cracks are considered to have occurred, as indicated in Figure 4b.

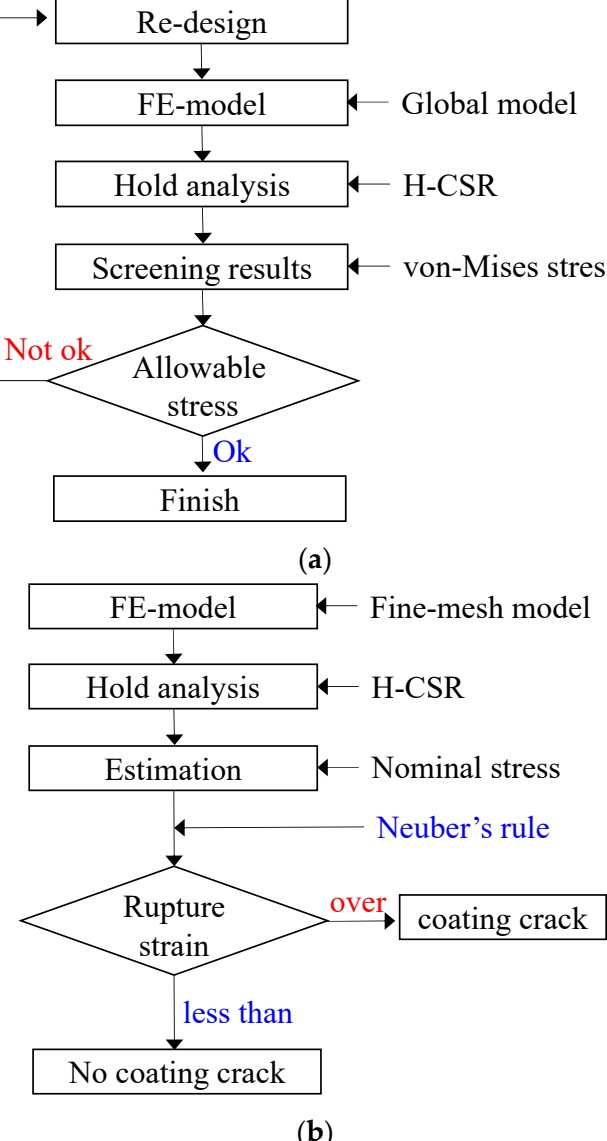

**Figure 4.** FE analysis procedure (global strength and coating strain). (**a**) global strength estimation. (**b**) coating strain estimation.

Based on structural analysis results, we selected a location with 70% or more allowable stress, assuming a primary possibility of coating crack occurrence under loading conditions; then detailed meshing was performed, as shown Figure 5. The mesh size is very small (about 1 mm) according to the coating thickness.

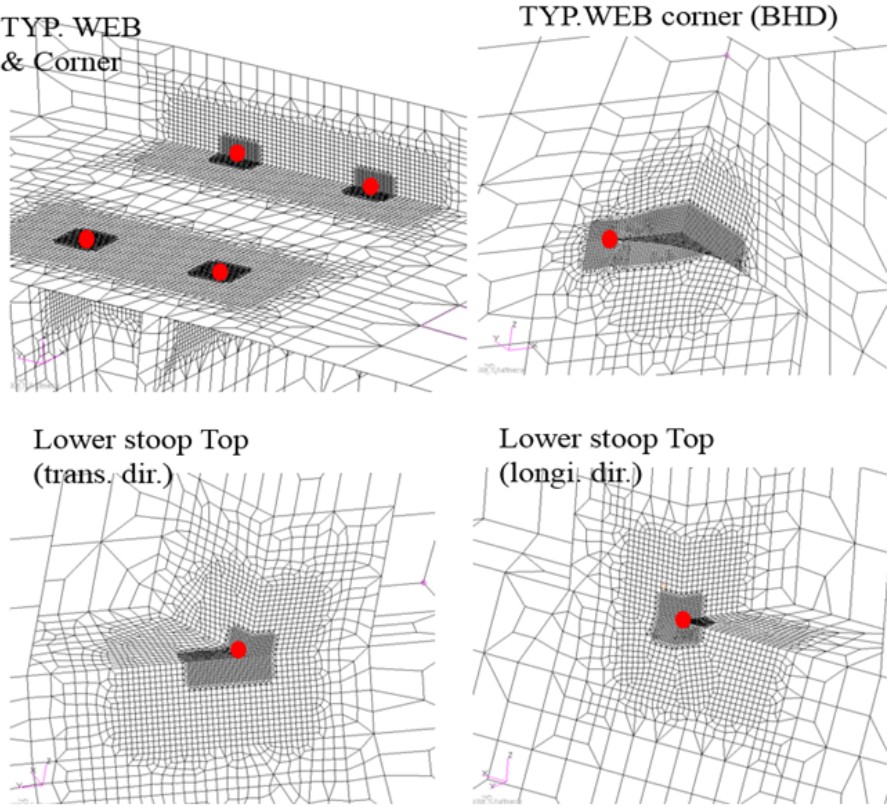

**Figure 5.** FE analysis modeling (critical area).

*3.2. Boundary and Load Conditions*

The boundary conditions are illustrated in Figure 6. Simply supported conditions were implemented at both ends of the FE model in order to simulate the continuous behavior of the hull structure against wave loads.

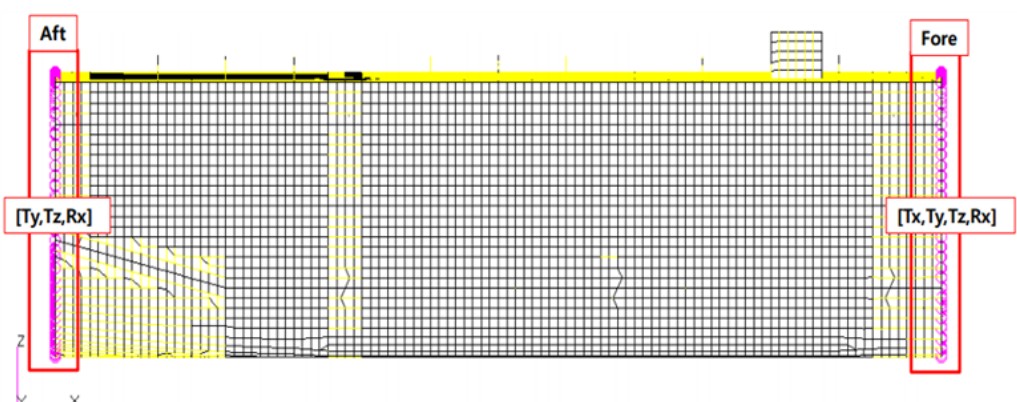

**Figure 6.** Boundary condition (elevation view).

There are two bending moments with respect to hogging and sagging for tank test conditions during sea trials based on the trim and stability data. Each moment is applied to the forward section, as shown Figure 7a,b. The cargo hold pressure and external pressure implemented during the tank test were also applied in the FE analysis, as shown in Figure 7c.

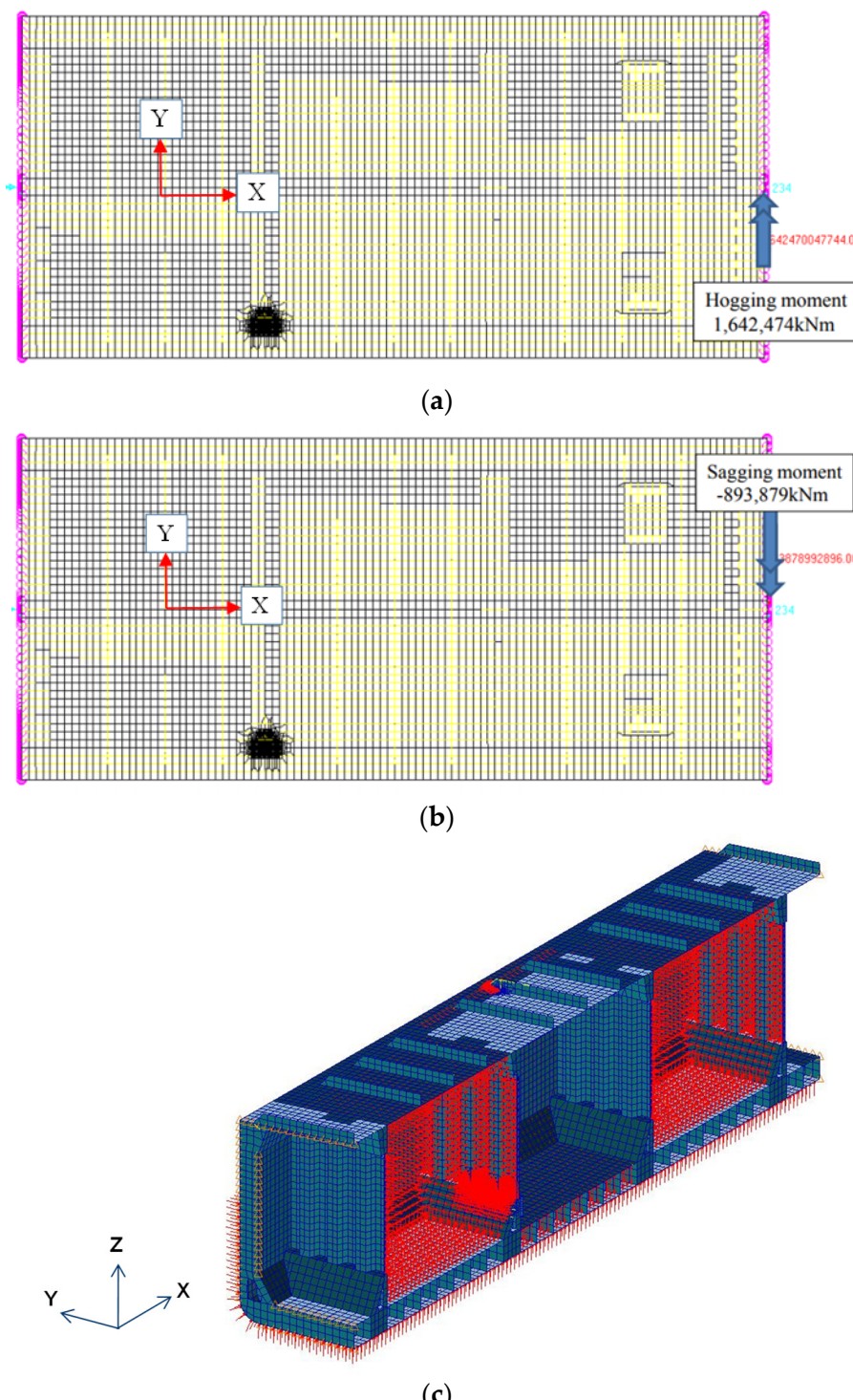

**Figure 7.** Load condition (plan view): (**a**) hogging application; (**b**) sagging application; and (**c**) internal/external pressure application.

### 3.3. Plastic Strain Calculation

The plasticity correction factor can be obtained from an actual cyclic stress–strain curve and Neuber's rules [7] or nonlinear finite element analysis, as shown Figure 8.

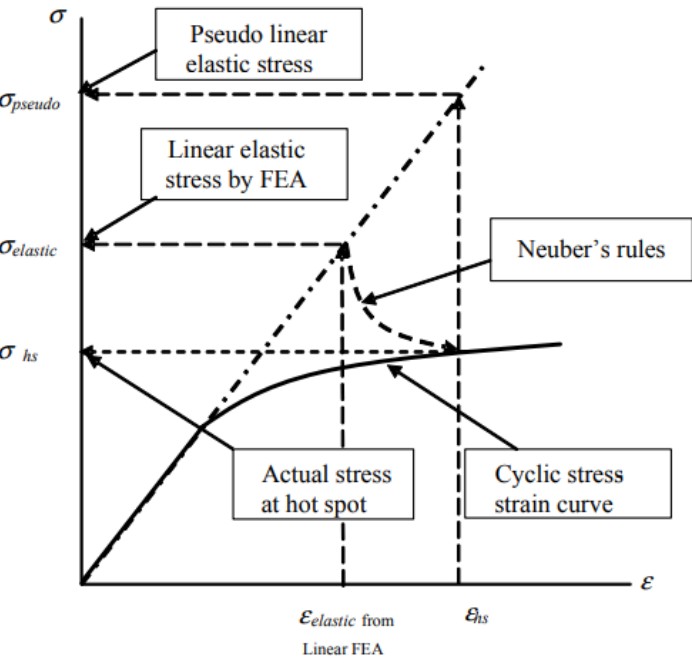

**Figure 8.** Definition of stresses and strain [7].

$$k_e = \frac{\sigma_{pseudo}}{\sigma_{elastic}} \tag{1}$$

where:

$\sigma_{elastic}$ = elastic hotspot stress obtained from linear elastic finite element analysis or a formula;

$\sigma_{elastic}$ = pseudo-linear elastic hotspot stress = $E \times \varepsilon_{hs}$.

In order to obtain the plasticity correction factor, a cycle stress–strain curve for materials should be obtained from experimental tests. Because it is difficult to obtain test data, it can be calculated through Equation (2).

$$\frac{\sigma_n^2 \times K^2}{E} = \frac{\sigma_{hs}^2}{E} + \sigma_{hs} \times \left(\frac{\sigma_{hs}}{K'}\right)^{1/n} \tag{2}$$

where:

$K$ = stress concentration factor;
$\sigma_{hs}$ = actual stress in the hot spot;
$\varepsilon_{hs}$ = actual strain in the hot spot;
$E$ = Young's modulus; and
$n$, $K'$ = material coefficient

### 3.4. Principal Stress Calculation

In order to calculate the nominal stress at the critical location using very fine meshing, an FE model should be developed. In general, structural safety is evaluated through global strength analysis using the membrane stress and ignoring the bending moment in the thickness, as shown Figure 9. To calculate the stresses and strains acting on the coating, it is necessary to confirm the results at the top of the plate, where maximum bending stress occurs, as shown Figure 9. In this case, the surface stress tends to be higher than membrane stress due to the addition of a bending moment.

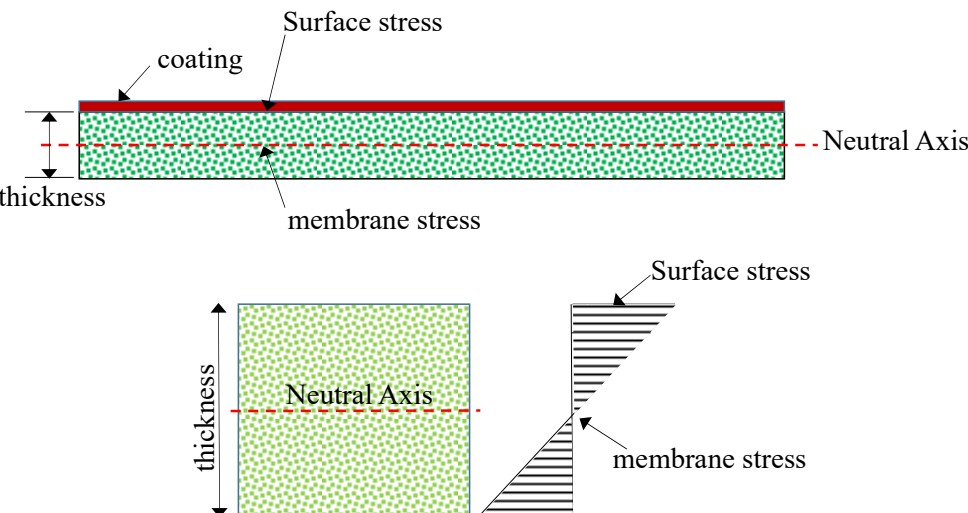

**Figure 9.** Definition of coating stresses and strain.

## 4. Results and Discussion

### 4.1. Global FE Analysis

The material of the structure was assumed as an isotropic finite element model, and the MSC Patran/Nastran 2012 analysis code was used. MSC Patran was used for pre/post-processing work to model the geometry, load, boundary conditions, and input material properties. MSC Nastran is based on complex and sophisticated numerical analysis methods and is the most advanced finite element solver. A total of 486,935 elements were used in the cargo hold model, with 940,045 nodes.

Figure 10 shows the global deformed shape of the middle cargo model under hogging conditions. The load and boundary conditions are well-described under the pure bending behavior in the FE model.

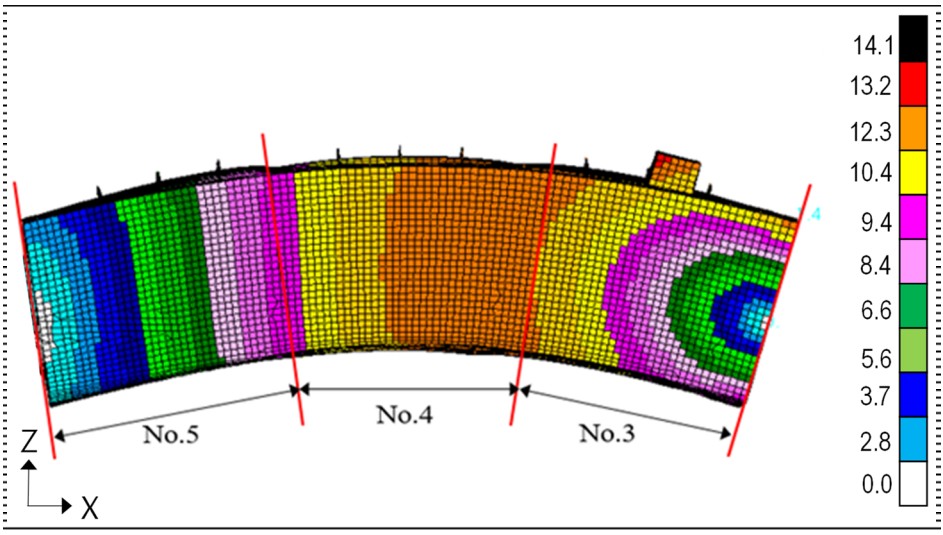

**Figure 10.** Deformed shape and displacement of three cargo hold models under hogging conditions (unit: mm).

In order to clarify the coating crack failure, we performed cargo hold analysis using coarse mesh modeling in order to identify a critical location that exceeds 70% of the allowable stress, as shown Figure 11.

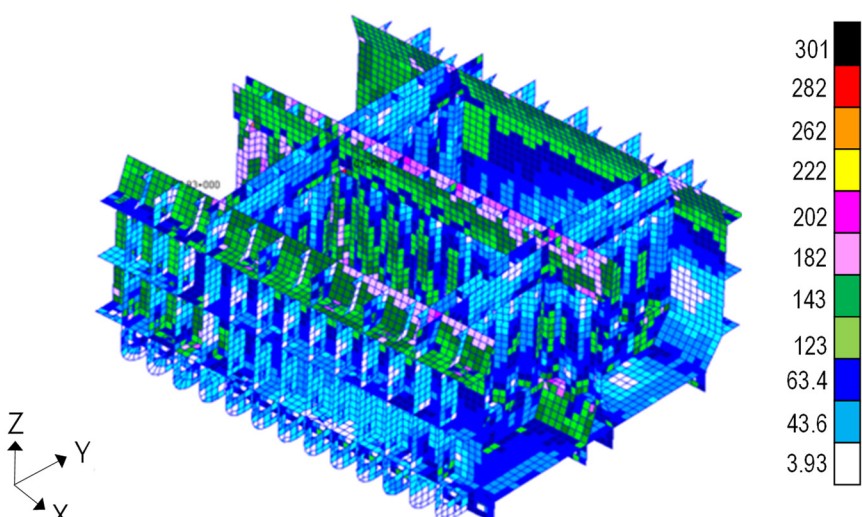

**Figure 11.** Von Mises stress contour of the inner cargo structure under hogging combinations.

In the cargo hold analysis, the stress value was verified in advance of determining the critical location, as illustrated in Figure 11. In particular, there are various connections in the corrugated bulkhead, where coating cracks often occur; therefore, the value should be carefully examined according to the geometric shape, as well as the location. Figure 12 shows a comparison of the stress results according to hull girder loads, namely hogging and sagging conditions. The maximum stress values are similar, but the stress patterns differ.

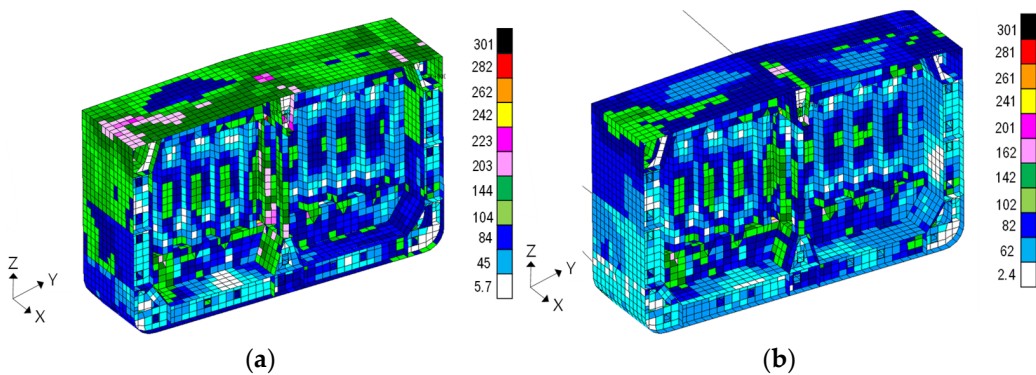

(**a**)                               (**b**)

**Figure 12.** Detailed stress contour between the center bulkhead with stools and inner structure in the cargo hold. (**a**) Hogging combination. (**b**) Sagging combination.

### 4.2. Acceptable Criteria

The results were verified against the acceptance criteria for all structural members within the longitudinal extent of the middle tanks of the FE model, as well as the regions forward and aft of the middle tanks up to the extent of the transverse bulkhead stringer and partial girder structure. The maximum permissible stress is based on the CSR (common structural rule) as defined by the mesh size and element type. The von Mises stress is calculated according to the membrane direction and shear stress of the element. Global strength analysis was performed to demonstrate that the permissible von Mises criteria specified in Table 1.

**Table 1.** Maximum permissible stress according to members.

| Structural Component | Yield Factor |
|---|---|
| Plate and primary supporting member | $1.0 \rightarrow (S + D)$<br>$0.8\ (S)$ |
| Corrugated bulkhead and supporting member | $0.9 \rightarrow (S + D)$<br>$0.7\ (S)$ |

*S*, static load; *D*, dynamic load component. A yield factor of 1.0 indicates that the allowable stress is considered up to the yield strength of the material.

*4.3. Detail FE Analysis for Coating Cracking*

Stresses at critical locations were calculated through structural strength FE analysis according to the CSR procedure [9,10]. The calculated principal stress can be recalculated by based on the plastic strain using Neuber's equation [7]. Plastic stress can be converted to principal stress using von Mises stresses at the critical locations where coating cracks are likely to occur. By analogy to the FE analysis results, 90% of the maximum von Mises stress was calculated as principal stress under tank test loading conditions and applied to the calculation of the plastic strain in Equation (2). In order to systematically classify the risk of the coating cracks according to location, risk was classified into three categories according to the magnitude of the strain rate, as shown Figure 13.

| HAZID Category | Level |
|---|---|
| Coating strain (%) > 1.00 | High RISK |
| 0.3 < Coating strain (%) ≤ 1.00 | Medium RISK |
| Coating strain (%) < 0.3 | Low RISK |

**Figure 13.** Risk categories of coating cracking according to strain rate.

Figure 14 shows the stress contour according to the locations determined by tank test loads. High-stress zones are generated at the joint between the bulkhead and the stool, the bracket connection end, and at the joint between the upper stool and the deck. The steel materials used for the cargo hold structures SS400 mild steel and AH32 and AH36 high-tensile steel. The allowable stress is applied variably (AH32, 489.5 MPa; AH36, 512.2 MPa) because the yield stress of the materials differs, according to the locations where coating cracking frequently occurs. According to the FE analysis results, plastic strain rates of coating were calculated for each location and summarized, as indicated Table 2. The coating strain is 0.3% or less at most locations, and the RISK index is in the low category. In other words, the RISK of damage to the coating is quite low under structural deformation conditions. Among the locations, the maximum coating strain of 0.75% occurred in the connection between the upper stool and the corrugated bulkhead. In the investigated target product carrier, no coating crack occurred due to structural deformation during the tank test. In this study, it was assumed that the coating was successfully installed without any defects.

**Table 2.** Summary of the coating strain and RISK index according to location.

| Structural Component | Coating Strain (%) | RISK Index |
|---|---|---|
| Deck transverse web | 0.33 | Medium |
| Lower stool–bracket | 0.40 | Medium |
| Longi. bottom shell | 0.11 | Low |
| Trans. corrugated BHD | 0.10 | Low |
| Upper stool–bracket | 0.41 | Medium |
| Upper stool bottom | 0.75 | Medium |
| Ceiling–upper stool top | 0.34 | Medium |

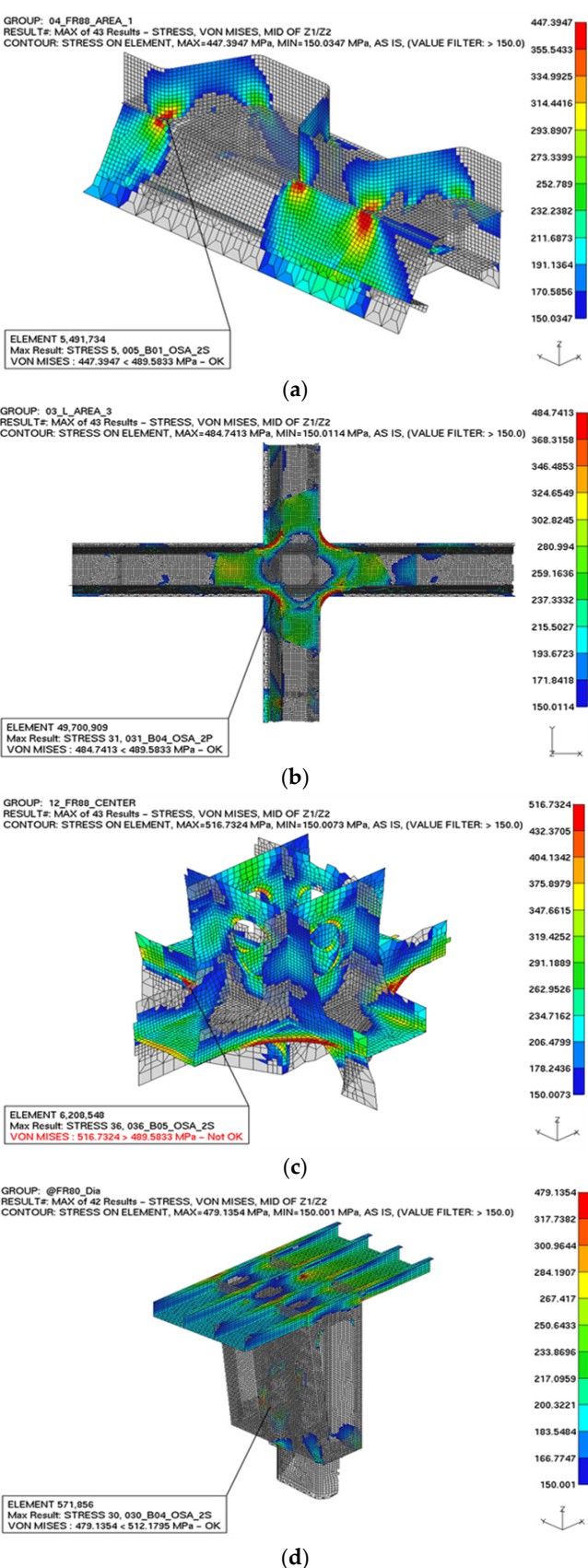

**Figure 14.** Maximum von Mises stress results with varying critical locations using a very fine meshing model: (**a**) lower stool–corrugated bulkhead connection; (**b**) center bulkhead–bracket connection; (**c**) upper stool–bracket connection; (**d**) upper stool–main deck connection.

#### 4.4. Alternatives for Anti-Coating Crack

During the commissioning process of the product carrier, a tank test is conducted to verify the safety of the structural members. In general, seawater is injected into the tank to the PORT (port side), the STBD (starboard side) is emptied, and the same pattern is maintained in the ballast tanks. These conditions represent a severe combination that is unlikely to occur under actual operating conditions of the vessel. Therefore, tank test procedure is adjusted to simulate the actual operating conditions and divided into a loading pattern. In addition, the difference of cargo density should be considered in the tank test process. In order to reduce the probability of coating cracking, the shape of the bracket is changed to a position of maximum stress from toe to center so hotspot stress can be dramatically reduced by increasing the weld toe, as indicated in Figure 15.

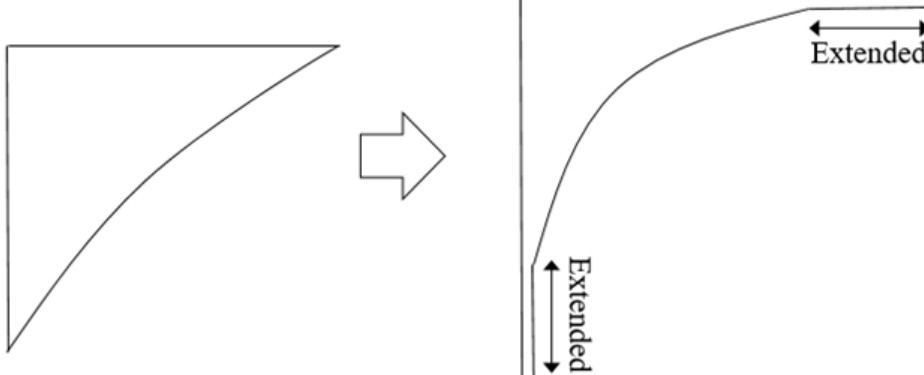

**Figure 15.** Corner bracket design change with extended toe.

The principal stress result is shown Figure 16a,b according to change in the bracket design. In case of a change in the shape at the weld toe, the stress at the center of the bracket increases by about 6%, whereas stress at the weld toe decreases by approximately 50%. Because the critical location where cracks occur in the coating is the end of the weld toe, modification of the shape is an effective method to reduce coating cracking.

An additional method of smooth grinding is applied on the welding toe, where high stress is concentrated because cracks occur more easily where the coating is thicker due to the characteristics of coating failure.

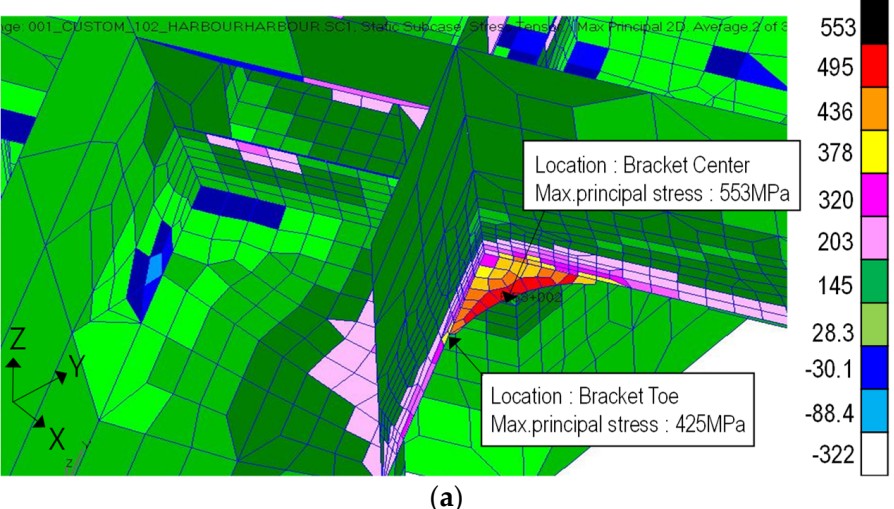

(**a**)

**Figure 16.** *Cont.*

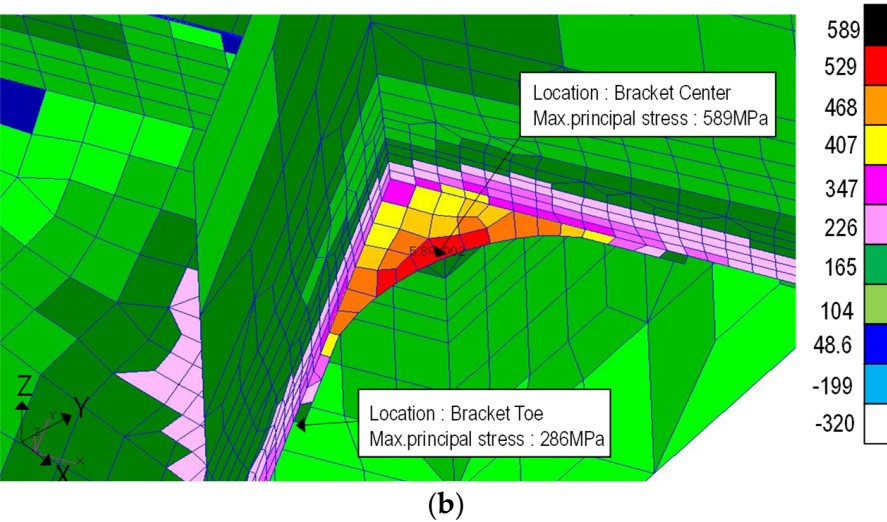

**(b)**

**Figure 16.** Principal stress result following a change in bracket design. (**a**) Maximum principal stress of the initial bracket model. (**b**) Maximum principal stress of the modified bracket model.

## 5. Conclusions and Future Works

In the present work, we investigated cases of coating cracks that occurred after tank tests under sea trial conditions. In order to clarify the mechanism of coating failure, numerical analysis was performed to verify the structural safety of coating failure under a combination of static and dynamic loads. Based on the obtained results, the following conclusions can be drawn.

(1). The coating strain must be applied with a very small mesh size; the first step should be calculation of the principal stress, which should then be converted into plastic strain at the critical locations;

(2). The governing load of coating cracking is caused by structural deformation, and cracks can be prevented when this phenomenon is accurately determined; and

(3). When the bracket shape is changed, the probability of coating cracking can be reduced; grinding of the weld bead is also helpful to prevent coating cracking.

In future, a database of coating cracks should be built using survey data from various ships in order to train an algorithm that predicts the crack location through a deep learning technique according to load and time.

**Author Contributions:** Conceptualization, J.-S.P. and K.-C.S.; methodology, J.-S.P. and M.-S.Y.; data curation, K.-C.S. writing—original draft preparation, K.-C.S.; writing—review and editing, M.-S.Y. All authors have read and agreed to the published version of the manuscript.

**Funding:** This research was supported by "Regional Innovation Strategy(RIS)"through the National Research Foundation of Korea(NRF) funded by the Ministry of Education(MOE) (2022RIS-002). This research was also supported by the Basic Science Research Program through the National Research Foundation of Korea (NRF) funded by the Ministry of Education(MOE) (NRF-2022R1I1A3068558).

**Data Availability Statement:** The data are not publicly available. The data presented in this study are available on request from the corresponding author.

**Conflicts of Interest:** The authors declare no conflict of interest.

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
