# Peer review of "Study on the Root Causes and Prevention of Coating Cracks in the Cargo Hold of a Product Carrier"

_metals, doi:10.3390/met12101688_

Round 1

Reviewer 1 Report

The author explores the coating cracking in the critical region caused by high stress in the cargo hold through numerical analysis, which has certain significance for designing the cargo hold of product carrier with lower structural stress. But some questions need to be answered so that readers can better understand and read.

1.    Tank Test and Coating Crack Survey

-The abbreviation “PORT” and “STBD” in Fig. 1(b) should be given the full name in the description of Figure 1, and this abbreviation appears again in the first paragraph of section 4.4 on page 11.

2. Results and Discussion

- The calculation results in Fig. 9, Fig. 10, Fig. 11 and Fig. 13 should be described in more detail, such as the relationship between the maximum von-Mises stress and the coating cracking..., which will help the reader's understanding.

- Do the No. 5 and No. 3 regions in Fig. 9 experience grid distortion during loading? The grids at the left and right borders and the upper and lower borders are significantly different.

- There should be a stress scale relative to the stress contour in Fig. 11.

-Why is it OK when the von-Mises stress<489.5833 MPa, but the critical value becomes 512.1795 MPa in Fig. 13(d)? Here the author should give a detailed description.

- The authors believe that by changing the stent shape, the structural stress can be reduced by about 50%, thereby reducing the cracking probability of the coating. This should give the result of the modified FE-model calculation (Fig. 14).

- Finally, and the most important question. Since the cargo hold model is large and the coating is relatively very small (about 1mm mentioned in section 3.1 of the manuscript), did the author use the global model or the component coating model as shown in Fig. 8? If the author adopts a global model, then the relationship between the stress maximum in the model and the cracking of the coating should be given. On the contrary, then at least a set of stress contour in the coating at the maximum stress location is given (any one of the maximum stress positions in Fig. 13 can be selected), and the corresponding analysis is given.

Author Response

Thank you for your comment. 

Please check the attached rebuttal document.

Reviewer 2 Report

The manuscript “Study on the Root Causes and Prevention of Coating Cracks in the Cargo Hold for Product Carrie”, by Myung-Su Yi, Kwang-Cheol Seo and Joo-Shin Park, highlights important calculus in the field of coating corrosion for cargo vessels. The manuscript can be accepted for publication, after minor corrections. My comments are below.

1.       Abstract: please reformulate it to be focused on what you studied in the manuscript; it is not clear your own contribution.

2.       Figure 8: what is the utility of the Membrane presented in figure 8? It is a dedicated material-type? Please insert some materials examples used here, because you are saying “coatings”, “membrane” stress, but it’s nothing inserted related to their material nature.

3.       In your simulations, have you considered some specific/ dedicated materials?

4.       Also, in your simulations, have you taken into account a high humidity factor? Because usually at the sea level and during storms, the humidity is very high, enhancing the high corrosion risk.

5.       Please add more references to your manuscript.

Author Response

Thank you for your comments.

Please check the attached rebuttal document.

Round 2

Reviewer 1 Report

The manuscript has been improved and the comments were fully addressed. It is recommended that the study be published in Metals.